

# Reliability of the weight-bearing ankle dorsiflexion range of motion measurement using a smartphone goniometer application

Helena Zunko[1] and Renata Vauhnik[2,3]

[1] Biotechnical Faculty, University of Ljubljana, Ljubljana, Slovenia, Ljubljana, Slovenia
[2] Faculty of Health Sciences, University of Ljubljana, Ljubljana, Slovenia, Ljubljana, Slovenia
[3] Arthron, Institute for Joints and Sport Injuries, Slovenia, Ljubljana, Slovenia

## ABSTRACT

**Background.** Weight-bearing ankle dorsiflexion range of motion measurement (weight-bearing lunge test) is gaining in popularity because it mimics lower extremity function in daily physical activities. The purpose of the study is to assess the intra-rater and the inter-rater reliability of the weight-bearing ankle dorsiflexion range of motion measurement with a flexed knee using a smartphone application Spirit Level Plus installed on an Android smartphone.

**Methods.** Thirty-two young, healthy subjects participated in the study and were measured in four sessions by two examiners. One measurement was taken on each ankle in every session. Eight measurements were taken from each participant. A total of 256 were taken from all the participants. The measurements for the individual subject were repeated no sooner than 24 hours after the first session. In order to assess the reliability, intraclass correlation coefficients (ICC), standard error measurements (SEM) and minimal detectable change (MDC) at the 95% confidence interval were calculated.

**Results.** Statistical data analysis revealed moderate intra-rater reliability for the right ankle (ICC = 0.72, 95% CI [0.49–0.85]) and good intra-rater reliability for the left ankle (ICC = 0.82, 95% CI [0.66–0.91]). Inter-rater reliability is moderate for the right (ICC = 0.73, 95% CI [0.52–0.86]) and the left ankle (ICC = 0.65, 95% CI [0.39–0.81]).

**Conclusion.** The observed method is moderately reliable and appropriate when the main objective is to assess ankle dorsiflexion mobility in weight-bearing when weight-bearing is not contraindicated. The concurrent validity of the Spirit Level Plus application is excellent.

Corresponding author
Renata Vauhnik,
renata.vauhnik@zf.uni-lj.si

## INTRODUCTION

Adequate ankle dorsiflexion range of motion (ROM) is necessary for performing daily physical activities like walking, rising from the sitting position, running and stair climbing (*Bohannon, Tiberio & Zito, 1997*; *Kluding & Santos, 2008*; *Konor et al., 2012*; *Rabin et*

*al., 2015*). Therefore, inadequate ankle dorsiflexion ROM directly affects individual's functioning, but also represents a risk factor for lower limb injuries (*Munteanu et al., 2009*), falls in the elderly (*Gehlsen & Whaley, 1990*) and development of other pathologies due to altered biomechanics and compensatory movement patterns (*Dinh et al., 2011*).

There is a wide variety of common causes of reduced ankle dorsiflexion ROM. The range may be reduced due to lower limb injuries such as ankle sprain (*Garrick & Requa, 1988*), the presence of other pathological conditions such as Achilles tendinopathy (*Duthon et al., 2011*), plantar fasciopathy (*Cheung, Zhang & An, 2006*), arthrosis (*Barg et al., 2013*), diabetes (*SearleMOsteo, Spink & Chuter, 2018*), post-stroke conditions (*Chung et al., 2004*), prolonged immobility (*Akeson et al., 1987*) and age-related changes in the mechanical properties of muscles and soft tissues (*Gajdosik, Van der Linden & Williams, 1999*).

Increasing ankle dorsiflexion ROM is a common physiotherapeutic goal. Current evidence of the effectiveness of therapeutic procedures is limited mainly by the absence of a uniform definition of physiological normative ankle dorsiflexion ROM values and differences in measurement procedures (*Charles, Scutter & Buckley, 2010*; *Young et al., 2013*). Valid, reliable, and accurate goniometric measurements are required in order to assess the effectiveness of therapeutic procedures and further treatment planning (*Jones et al., 2005*; *Wilken et al., 2011*). Weight-bearing measurement is gaining popularity and is more suitable for function assessment than non weight-bearing ankle dorsiflexion ROM measurement procedures (*Powden, Hoch & Hoch, 2015*; *Zunko & Puh, 2016*). The procedure mimics a functional position of the lower limb during daily physical activities, hence the term functional measurement has been proposed by some authors (*Jones et al., 2005*; *Krause et al., 2011*; *Dickson et al., 2012*; *Rabin & Kozol, 2012*).

Several different measurement procedures have been developed using different measurement tools. The most commonly used measurement tools are different types of goniometers and tape measure. Some authors used special devices (*Jones et al., 2005*; *Watson, Boland & Refshauge, 2008*; *Morales et al., 2017*; *Munteanu et al., 2009*) or mobile applications of goniometers (*Banwell et al., 2019*; *Gosse et al., 2021*; *Williams, Caserta & Haines, 2013*; *Vohralik et al., 2015*).

In medicine, there is a growing trend of using various mobile applications on smartphones instead of standard measuring tools because of their accessibility, affordability and simplicity (*Franko & Tirrell, 2012*). *Williams, Caserta & Haines (2013)*, who measured ankle dorsiflexion ROM in weight-bearing using the mobile goniometer application TiltMeter app installed on a smartphone (IOS operating system), reported that the measurement intra-rater and inter-rater reliability with the knee extended and flexed is excellent (ICC 0.8 or more). Two other studies (*Gosse et al., 2021*; *Banwell et al., 2019*), investigating the use of Apple IOS based goniometer applications for weight-bearing ankle dorsiflexion ROM measurement, determined moderate to excellent reliability and excellent validity.

Although there are several studies, reporting the reliability of the weight-bearing ankle dorsiflexion ROM measurement using Apple IOS based goniometer applications, evidence of the reliability of a mobile goniometer application using an Android smart phone for weight-bearing ankle dorsiflexion ROM measurement is limited. Therefore, the aim of

our study was to determine inter-rater and intra-rater reliability of the weight-bearing ankle dorsiflexion ROM measurement with the flexed knee, using the mobile goniometer application Spirit Level Plus installed on an Android smartphone.

## MATERIALS & METHODS

This study followed a test-retest design to determine intra-rater and inter-rater reliability of the weight-bearing ankle dorsiflexion ROM measured by an experienced physiotherapist and a physiotherapy student. The concurrent validity for the Spirit Level Plus application was established when compared to the universal goniometer at angle 0 and 45 degree angles.

The study was approved by the Republic of Slovenia National Medical Ethics Committee (No. 0120-235 / 2017/5).

### Participants

The inclusion criteria for participation in the study was absence of musculoskeletal injuries in the lower limbs or other disorders of the neuromuscular system, including inflammatory joint conditions, in the last six months prior to the measurements. 32 young, healthy participants (23 women (72%) and nine men (28%), age 20.9 ± 1.7 years, body height 170.7 ± 8.1 cm, body mass 66.2 ± 11.6 kg), who signed an informed consent, participated in the study. Research recruitment leaflets were used for participants recruitment. Descriptive statistics of the participants are presented in Table 1.

### Examiners

The measurements were conducted by a physiotherapist with 16 years of clinical experience in the field of musculoskeletal physiotherapy, who has been using this measurement procedure regularly, and a final year physiotherapy student who has been instructed how to perform the measurements prior to the study. Prior to the main data collection, the pilot study was conducted to refine the protocol. Both examiners are right-handed.

### Measurement of the weight-bearing ankle dorsiflexion ROM

The measurement procedure was demonstrated to all participants one-on-one. The first one to conduct the measurements were the experienced physiotherapist, followed by the physiotherapy student. At least 24 h elapsed between measurements in each participant. Eight measurements were taken from each participant. A total of 256 measurements were taken from all the participants. Examiners performed the first and the second measurements on different days, therefore the stretch from the first measurement could not influence the result of the second measurement. Left and right ankle dorsiflexion ROM were measured, the left ankle dorsiflexion ROM was measured from the participant's left side, and the right ankle dorsiflexion ROM from the right side. Due to a rather large sample size, and the fact that both sides were being measured, it was impossible for the examiners to memorise the results. The measurements were collected on paper separately, so the examiners were blinded to each other's measurements.

The participants were requested to stand in front of the wall and they were allowed to use the wall for support if needed. They were then asked to take one step back with the leg

**Table 1 Descriptive statistics of the participants.**

|  | Age (years) | Height (cm) | Body mass (kg) |
|---|---|---|---|
| Mean | 20.9 | 170.7 | 66.2 |
| Median | 21 | 169.5 | 64 |
| Standard deviation | 1.7 | 8.1 | 11.6 |
| Range | 19–27 | 155–187 | 50–95 |

that was measured and place the foot parallel to the other in the direction perpendicular to the wall. The next instruction was to move the knee forward toward the wall, aligned over the second toe, and stop just before the heel starts lifting off the ground. At this point the measurements were taken by placing the short side of the smartphone (Huawei P8lite) on the posterior part of the Achilles tendon, one centimetre above the posterior calcaneal uberosity while using the mobile goniometer application Spirit Level Plus to measure the tibia inclination relative to the floor (Fig. 1). The application Spirit Level Plus (now named Spirit Level) was developed by Keuwlsoft (http://www.Keuwl.com). The application is free to download, no subscription is needed. An Android 1.6+ operating system is required. Prior to the measurement, the smartphone was placed with its long axis on the floor and calibrated to 0°. A similar procedure has been used by other researchers (*Bennell et al., 1999*; *Burns & Crosbie, 2005*; *Rose, Burns & North, 2010*; *Williams, Caserta & Haines, 2013*; *Banwell et al., 2019*; *Gosse et al., 2021*).

The examiners were supervising the movement of the heel and the knee by holding the heel and guiding the movement of the knee while the subject moved it forward towards the wall. If the heel started to lift, the procedure was stopped and repeated. All the measurements were performed while the subjects were barefoot.

## Statistical procedure

Reliability of measurements was assessed by intraclass correlation coefficients (ICC) and standard errors of measurement (SEM) from which minimum detectable change (MDC) was determined. ICC (2,1) was used to calculate (a) intra-rater as well as (b) inter-rater reliability.

The degree of reliability of the test, measured by ICC, was determined according to the classification of *Portney, Watkins et al. (2009)* (value between 0.00–0.49 denotes poor, 0.50–0.79 moderate and 0.80–1 good reliability). Concurrent validity of the application was explored using ICCs (Model 3,1) (Two-way mixed effect with absolute agreement). The R statistical program (*R Core Team, 2013*) was used to analyse the data. The level of statistical significance was set to alpha = 0.05.

## RESULTS

Ankle dorsiflexion ROM measurements are presented in Table 2.

Intra-rater reliability was good (and at least moderate when considering the confidence) for the left leg (ICC = 0.82, 95% CI [0.66–0.91]) and moderate for the right leg (ICC = 0.72, 95% CI [0.49–0.85]). SEM was less than 1.9 degrees. Minimum detectable change

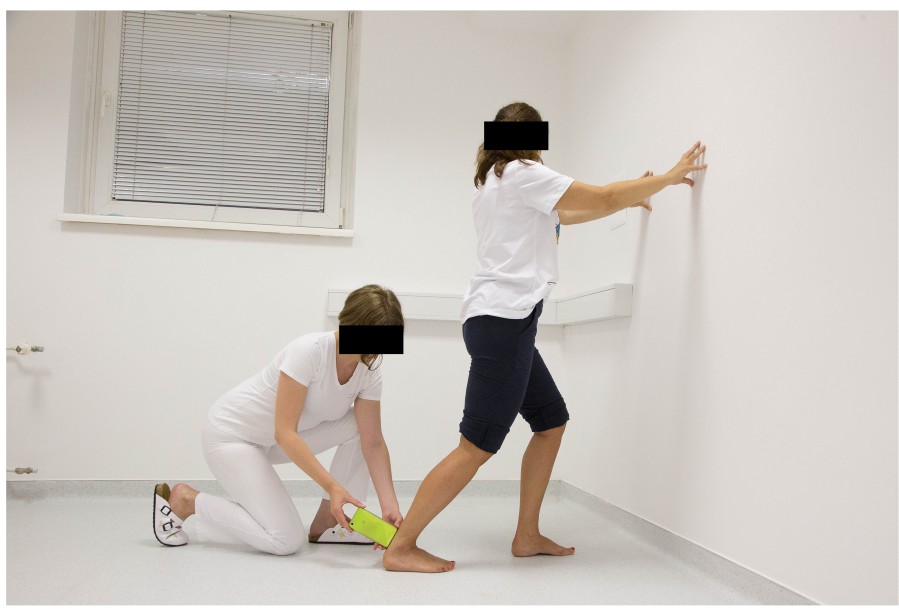

**Figure 1 Weight-bearing ankle dorsiflexion range of motion measurement.**

**Table 2 Ankle dorsiflexion range of motion measurements.**

| | Examiner 1 | | | | Examiner 2 | |
| | Measurement 1 (°) | | Measurement 2 (°) | | Measurement 1 (°) | |
| | Right | Left | Right | Left | Right | Left |
|---|---|---|---|---|---|---|
| Mean | 31.6 | 31.4 | 30.7 | 30.8 | 31.2 | 30.2 |
| Standard deviation | 3.3 | 4.2 | 3.6 | 4.1 | 3.8 | 3.4 |
| Range | 25–38 | 25–41 | 23–42 | 23–39 | 25–40 | 23–37 |

**Table 3 Intra-rater and inter-rater reliability.**

| | ICC | 95% IC | SEM | MDC |
|---|---|---|---|---|
| Intra-rater reliability | | | | |
| Right | 0.72 | 0.49–0.85 | 1.89 | 5.24 |
| Left | 0.82 | 0.66–0.91 | 1.80 | 4.99 |
| Inter-rater reliability | | | | |
| Right | 0.73 | 0.52–0.86 | 1.89 | 5.23 |
| Left | 0.65 | 0.39–0.81 | 2.34 | 6.49 |

Notes.

Abbreviantions: ICC, intraclass correlation coefficient; CI, confidence interval; SEM, standard error of measurement; SEM, $SD \times \sqrt{1-ICC}$; MDC, minimum detectable change (based on 95% CI); MDC, $1,96 \times SEM \times \sqrt{2}$.

was determined at least 4.99 degrees. The results are shown in Table 3 and Bland-Altman plots are presented in Fig. 2.

Inter-rater reliability is moderate for the right leg (ICC = 0.73, 95% CI [0.52–0.86]) and slightly poorer for the left leg (ICC = 0.65, 95% CI [0.39–0.81]) topped 2 degrees.

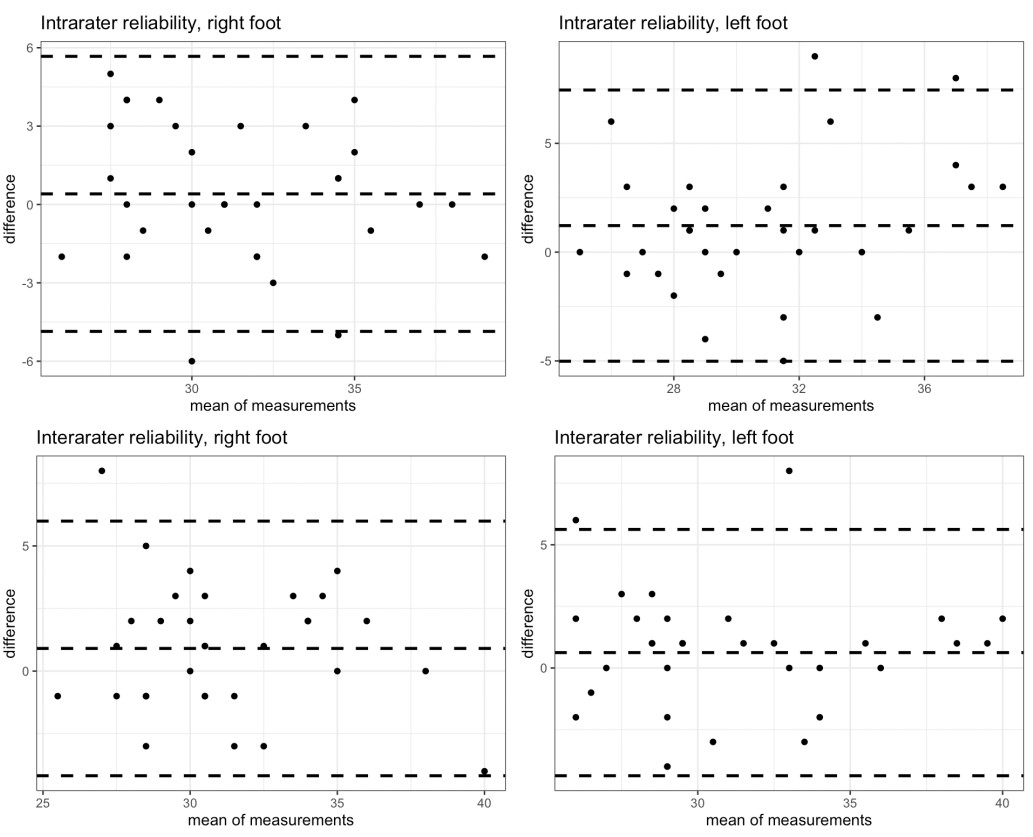

**Figure 2 Bland-Altman plots.**

SEM was slightly larger than within intra-rater reliability and topped at 2.34 degrees. The smallest detectable change in this group of subjects was determined to be at least 5 degrees. The concurrent validity of the Spirit Level Plus application is excellent, indicated by an ICC >0.999, with 95% confidence interval of [0. 999822, 0. 9999622]. The Bland-Altman plot for concurrent validity is presented in Fig. 3.

## DISCUSSION

The intra-rater and inter-rater reliability of the weight-bearing ankle dorsiflexion ROM measurement was moderate to good with ICC ranging from 0.65 to 0.85. The results are comparable to the results of *Gosse et al. (2021)*, who used the mobile determined moderate to excellent reliability of the mobile goniometer application iPhone level, with ICC ranging from 0.68 to 0.90. However, the results are only partially comparable to the results of *Williams, Caserta & Haines (2013)*, who used mobile goniometer application TiltMeter app on an Apple iPhone and have reported good reliability (ICC 0.8 or more). There are several possible explanations for why their reliability is higher and the most obvious one is that *Williams, Caserta & Haines (2013)* were using a different mobile goniometer application on a different type of device with a different operating system (Android *versus* iOS). Differences in measurements could also be influenced by the application

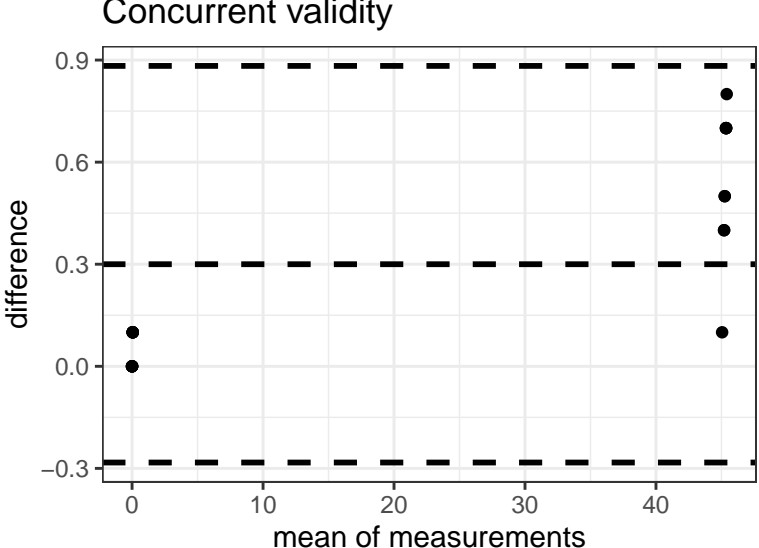

**Figure 3** **Bland-Altman plot of the concurrent validity.**

itself. Applications and their specific software platforms need to be properly validated, and any available new version of the application should be re-validated (*Mobile Medical Applications - Guidance for Industry and Food and Drug Administration Staff, 2013*). The concurrent validity of the Spirit Level Plus application is excellent >0.999, with 95% confidence interval of [0. 999822, 0. 9999622]. The Bland-Altman plot (Fig. 3) shows a tendency to larger measured angles with the Spirit Level Plus application. The tilt is larger with the large angle, however it stays within 1 degree. The mean difference between the two methods is 0.3 degrees with 95% confidence interval of [−0.28, 0.88].

There were some differences in the measurement procedures used in the present study as compared to *Williams, Caserta & Haines (2013)*. While *Williams, Caserta & Haines (2013)* measured only the right ankle, we measured both ankles. The left was measured from the participant's left side, and the right one from the right side, so the examiner's dominant hand didn't always have the same function. This could explain the differences in the ICC between the left and right side in our subjects.

Inter-rater reliability might be lower in the present study due to the inexperience of the physiotherapy student. A novice rater that participated in the study of *Williams, Caserta & Haines (2013)*, already had 2 years of clinical experience and had routinely used the weightbearing ankle dorsiflexion ROM measurement technique during clinical examinations as well as the experienced rater. Interestingly, the intra-rater reliability of a clinician in training and a novice rater was higher (ICC 0.90) when using the iPhone app than the intra-rater reliability of an experienced clinician (ICC 0.75) (*Gosse et al., 2021*).

It has been suggested that, rather than calibrating the device on the floor, it might be more appropriate to calibrate the device when it is placed on the Achilles tendon while the subject is standing in a neutral starting position. By doing so we would measure the actual range of motion. Another suggestion was, that it might be more suitable to use the longer

side of the smartphone, rather than the short one, but because of the different calf shapes we doubt that this argument has much validity. Another limitation of this study was that participants were not gender balanced.

In general, the limitation of the weight-bearing ankle dorsiflexion ROM measurements is that it cannot be used when weight-bearing is contraindicated (*Bennell et al., 1998*; *Konor et al., 2012*; *Rabin & Kozol, 2012*). The procedure is less objective than the non-weight-bearing passive one, because the subject is the one that determines the load applied to reach the end position. On that account the examiner also loses important information about the end feel of the movement (*Zunko & Puh, 2016*). On the other hand, the procedure is fast, simple and is more suitable to assess functional ankle dorsiflexion ROM (*Konor et al., 2012*; *Rabin & Kozol, 2012*; *O'Shea & Grafton, 2013*). Unlike passive non-weight-bearing measurement of ankle dorsiflexion ROM it can easily be performed by one examiner only, which is an important advantage for the examiners that must perform the measurements alone due to different circumstances (*Palmer & Epler, 1990*; *Jakovljević & Hlebš, 2011*).

One advantage of using the mobile application rather than the universal goniometer is that determining the axis of rotation is not necessary. This also applies to the digital and gravity goniometers. *Konor et al. (2012)* simplified weight-bearing ankle dorsiflexion ROM measurement using the universal goniometer by aligning the stable branch of the goniometer with the floor and not the fifth metatarsal as performed by *Dickson et al. (2012)*.

*Rabin & Kozol (2012)* recommend choosing the measurement method according to the aim of the measurement. If the aim is to primarily assess individual's functioning (walking, stair climbing, squatting etc.), detectable in the weight-bearing method is more appropriate. By choosing this method, we can also avoid false negative results, since in some individuals the difference in ankle dorsiflexion ADROM between the affected and the non-affected side is only detectable in weight-bearing position (*Jones et al., 2005*). The correlation between weight-bearing ankle dorsiflexion ROM and non-weight-bearing ADROM is only moderate due to three to four times higher forces affecting the joints of the foot during standing (*Jones et al., 2005*; *Krause et al., 2011*).

Differences in measuring procedures could also affect the results of the measurements (*Krause et al., 2011*; *Rabin & Kozol, 2012*). Examiners used various measuring tools for the weight-bearing ankle dorsiflexion ROM measurements (universal classical goniometer, liquid or digital gravity goniometer, mobile applications of goniometers, centimetre measuring tape, ruler or special devices), some performed measurements on the front of the leg, others on the rear of the leg, with the knee flexed or extended.

Those who used a gravity goniometer or a mobile goniometer application measured the tibial inclination on different locations (superior to the posterior calcaneal tuberosity (*Dickson et al., 2012*; *Williams, Caserta & Haines, 2013*; *Banwell et al., 2019*; *Gosse et al., 2021*), on the lateral (*Cejudo et al., 2014*) or anterior (*Bennell et al., 1998*; *Dickson et al., 2012*; *Vohralik et al., 2015*) part of the tibia, at different heights). Those who used a universal classical goniometer placed the fixed arm parallel to the fifth metatarsal or the ground. *Dickson et al. (2012)* suggested liquid gravity goniometer, placed superior to the posterior calcaneal tuberosity, as the most appropriate measurement tool for the weight-bearing ankle dorsiflexion ROM measurement. Several authors used digital

gravity goniometer (*Munteanu et al., 2009*; *Krause et al., 2011*; *Evans, Rome & Peet, 2012*; *Konor et al., 2012*; *Williams, Caserta & Haines, 2013*; *Banwell et al., 2019*; *Gosse et al., 2021*). Mobile goniometer applications, based on built-in sensors, have similar features as digital and liquid gravity goniometers (*Williams, Caserta & Haines, 2013*; *Vohralik et al., 2015*). Affordability is one of the advantages of mobile applications, as many of them are cost-free. *Williams, Caserta & Haines (2013)* used Tiltmeter, a mobile application installed on an IOS smartphone in their study, but there are many applications available that can be used on Android smartphones as well (*Mourcou et al., 2015*). In our study we used the application Spirit Level Plus on an Android Huawei P8lite smartphone, indicating at least moderate reliability.

Minimal detectable change in our study was higher (5.0°–6.5°) as compared to the results of *Williams, Caserta & Haines (2013)* (2.2°–4.4°), *Gosse et al. (2021)* (2.10°–5.7°) and *Banwell et al. (2019)* (2.4°–5.0°). In the studies where universal classical goniometer was used to measure ankle dorsiflexion ROM in the weight-bearing position (*Konor et al., 2012*; *Dickson et al., 2012*), minimal detectable change was 5° to 7.7°, which does not deviate significantly from our results, nor is the difference clinically significant. A literature review conducted by *Zunko & Puh (2016)* revealed that minimal detectable change was smallest in studies where they used liquid gravity goniometers (1.5°–3.9°) (*Bennell et al., 1998*; *Dickson et al., 2012*; *Cejudo et al., 2014*) or mobile goniometer applications (2.2°–5.2°) (*Williams, Caserta & Haines, 2013*). In the study by *Konor et al. (2012)*, *Cejudo et al. (2014)* and our study, reliability was evaluated for both legs, and in the study by *Williams, Caserta & Haines (2013)* reliability was performed only for the right leg and in study by *Dickson et al. (2012)* and *Bennell et al. (1998)* reliability was performed only for the left leg.

## CONCLUSIONS

Inter-rater and intra-rater reliability of the weight-bearing ankle dorsiflexion ROM measurement using the mobile goniometer application Spirit Level Plus installed on an Android smartphone is moderate. The concurrent validity of the Spirit Level Plus application is excellent. Further work is required to determine the normative values of ankle dorsiflexion ROM and to determine the differences in ankle dorsiflexion ROM among different age groups and gender before its recommendation in clinical settings.

## ACKNOWLEDGEMENTS

We thank the physiotherapy student for their work as a second examiner and to all the participants for their cooperation.

### Funding

This work was supported by the Slovenian Research Agency (research core funding no. P3-0388). The funders had no role in study design, data collection and analysis, decision to publish, or preparation of the manuscript.

## Grant Disclosures

The following grant information was disclosed by the authors:
Slovenian Research Agency: P3-0388.

## Competing Interests

The authors declare there are no competing interests.

## Author Contributions

- Helena Zunko conceived and designed the experiments, performed the experiments, analyzed the data, prepared figures and/or tables, authored or reviewed drafts of the paper, and approved the final draft.
- Renata Vauhnik conceived and designed the experiments, authored or reviewed drafts of the paper, and approved the final draft.

## Human Ethics

The following information was supplied relating to ethical approvals (i.e., approving body and any reference numbers):

The study was approved by the Republic of Slovenia National Medical Ethics Committee (No. 0120-235 / 2017/5).

## Data Availability

The raw measurements are available in the Supplemental File.

## Supplemental Information

Supplemental information for this article can be found online at http://dx.doi.org/10.7717/peerj.11977#supplemental-information.

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
