# Peer review of "Reliability of the weight-bearing ankle dorsiflexion range of motion measurement using a smartphone goniometer application"

_PeerJ, doi:10.7717/peerj.11977_

## Round 0.1 · original submission · Major Revisions

We received three consistent reviews for the paper. Although reviewers identified some merits of the paper, there are a number of issues that need to be rectified. Note that reviewer recommended references should only be cited if they are essential. Be sure that the ignorance of unjustified references won’t affect the final outcome of reviews on the revision.

·

Basic reporting

The reporting is clear but insufficient in its use of existing evidence in some areas. I have covered this more completely in the last section.

Experimental design

The research is well within the Aims and Scope of the journal of submission. The research question is well defined. There are some concerns with how it rationales the 'gap' it fills by not referring to current evidence in the field. The method of investigation needs further explanation but maybe appropriate if described more fully.

Validity of the findings

The findings add to a body of knowledge on the use of smartphones. Further information is needed in the methods to determine the reach or interest it may have. Data has been provided. Analysis requires some attention but what has been completed appears robust. Conclusion is sound.

Additional comments

Thank you for the opportunity to review the manuscript “Reliability of the functional ankle dorsiflexion range of motion measurement in young adults”. The article is well written, however there are some concerns that are prohibiting the recommendation for publication at this point.
Major concern: the measure capabilities of the app are not being compared to standard practice or ‘gold standard’ measures (e.g. analogue or digital inclinometers). The reliability of the app is not in question but the authors need to be very clear that this does not guarantee the outcomes are valid. The app could consistently measure incorrectly. Determining validity of the app against a known standard is strongly recommended prior to continuing for publication. This can be done comparing outcomes of the app to known angles (e.g. compare the app to a digital inclinometer and 0 degrees, 15 degrees and 45 degrees using solid and immovable reference points, such as inclinations on ramps).
Second recommendation would be to reduce the use of abbreviations, particularly non-standardised abbreviations such as ADROM. It introduces a risk of confusion, breaks sentence flow for the reader and rarely has an adequate effect on word count to be of value. I would recommend the abbreviation be replaced with ankle dorsiflexion ROM throughout. Furthermore, it may be prudent to identify the measure as weight bearing (e.g. weight-bearing ankle dorsiflexion ROM).
Final issue is that much of the literature cited in the background and discussion is > 5 years old. Many newer publications review Android and IOS apps in adults and children. It would be good to update some references, particularly as the Tilt-meter app, which is mentioned several times throughout the manuscript, was decommissioned in 2018.

Abstract
including the number of measures made and more information on the participant cohort would benefit.

Background
Good rationale for the need to measure weight-bearing ankle dorsiflexion ROM is given. Further breadth of referencing could be done for the last section where it is currently limited to Williams et al., for using IOS based measures given many other studies have investigated ankle dorsiflexion – e.g:
IOS measures of weight bearing ankle dorsiflexion:
Banwell, H.A., Uden, H., Marshall, N. et al. The iPhone Measure app level function as a measuring device for the weight bearing lunge test in adults: a reliability study. J Foot Ankle Res 12, 37 (2019). https://doi.org/10.1186/s13047-019-0347-9
Carlos Balsalobre-Fernández, Natalia Romero-Franco & Pedro Jiménez-Reyes (2019) Concurrent validity and reliability of an iPhone app for the measurement of ankle dorsiflexion and inter-limb asymmetries, Journal of Sports Sciences, 37:3, 249-253, DOI: 10.1080/02640414.2018.1494908
This section would also benefit from further rationale on why targeting android apps is important. It erroneously states that there are no publications reporting reliability of Android smart phone apps, which is incorrect (see below) – however the argument could be make that the majority of papers published within the last 5 years are all using Apple IOS based goniometers, and those using Android apps are not measuring dorsiflexion, e.g:
Motaz Abdalla Alawna, Bayram H. Unver, Ertugrul O. Yuksel; The Reliability of a Smartphone Goniometer Application Compared With a Traditional Goniometer for Measuring Ankle Joint Range of Motion. J Am Podiatr Med Assoc 1 January 2019; 109 (1): 22–29. doi: https://doi.org/10.7547/16-128

Methods
Could the authors begin this section with a study design description.
Participants: It would be good to include the age, height and body weight range of participants and offer some indication of where they were recruited from (e.g. a convenience sample of university students) and how recruitment was conducted..
Please indicate if inflammatory joint conditions were also an exclusion criteria.
Examiners: Raters need to be explained further. Does the experienced physiotherapist use this measure frequently in practice and had the final year physiotherapy student being instructed on how it was conducted prior to the study? Did they pilot the study first and refine the protocol?
Measurement of the functional ADROM: How was the measurement demonstrated to all participants, as a group on one on one.
Given the physiotherapist measured first each time, does this introduce potential order effects (e.g. increasing available ROM due to stretch from the first measure)?
Can the authors describe the timing and protocol between raters, e.g. could the person rest between? And how were the measures collected (on paper?).
Did the experienced physiotherapist complete and record measures in the presence of the student (or vice versa)?
I also question if 32 people is a large enough sample to ensure that raters don’t have memory of previous results. Also, can the authors discuss the limitations of ‘double dipping’ when you use both legs on the one person (Menz, 2004)
Menz, H. (2004). Two feet, or one person? Problems associated with statistical analysis of paired data in foot and ankle medicine. Foot, 14, 2 - 5.
Further information on the App, e.g. reference to its developers and which operating systems are required etc., and is it free to download or involve a subscription etc., .
Statistical procedure
How was all the data managed? It would benefit to include the participant characteristic management here (e.g. Descriptive statistics were used to describe participants…).
Can the authors rationalise the use of ICC random rather than two-way mixed effect (i.e. Model 2, 1 rather than Model 3, 1)?
The authors could consider using Bland Altman Plots to ensure the limits of agreement of raters is within an acceptable level. This could also be used to explore validity if the app is compared with a digital goniometer as recommended.

Results
Please change the word smallest to minimum in line 134.
Discussion
You could incorporate this paper into your discussion given it included a final year physiotherapy student and a novice (parent). It would offer you a clearer comparison that the Williams study.
Gosse G, Ward E, McIntyre A, Banwell HA. 2021. The reliability and validity of the weight-bearing lunge test in a Congenital Talipes Equinovarus population (CTEV) PeerJ 9:e10253 https://doi.org/10.7717/peerj.10253
Sentence beginning line 206 requires a reference.
Please replace comma’s for full stops when reporting the Minimal Detectable change (Lines 220 to 228).

References:
Evans, Rome & Peet, 2012 is not in the reference list. Please review to ensure all other references are included.

Reviewer 2 ·

Basic reporting

The manuscript was very clear, easy to read and was conducted well. Considering the language, I just found few suggestions and typing errors. In lines 79-83 (at the end of introduction) I would suggest to make two different sentences instead of one, and in the heading of table 2 there is a small spelling mistake.
In the Table 1, the Range is somewhat confusing, I would suggest to leave that out or show the range more understandable, like 19-27.

Experimental design

Fine, no comments to add.

Validity of the findings

Interesting study, no comments to add.

Reviewer 3 ·

Basic reporting

no comment

Experimental design

Some comments in general section

Validity of the findings

no comment

Additional comments

I have read the article ID 58796 under title “Reliability of the functional ankle dorsiflexion range of motion measurement in young adults”.


In this article, the authors reported their work using a mobile goniometer application Spirit Level Plus from Android smartphone and recruited 32 subjects in study. The overall, article writing is good and the results analyses are reasonably thorough. But although, the authors used new application to measure the ankle dorsiflexion, the study is still preferable that if the authors can take the following comments into consideration in preparing the final version.

I recommend that the authors make changes related to the comments below.




Reviewer comments:

-Figure 1, shown (weight bearing lunge test) but at the same time what the degree of knee flexion during test. Maybe the same subject when did re-test the degree of knee is different so I think that the best position is for the knee to be parallel to the foot, and the distance (to the nearest 0.lcm) from the end of the big toe to the wall using a tape measure on the floor and, for more information could see references:

- In line 297(Konor et al.);
- Bennell et al, 1998. Intra-rater and inter-rater reliability of a weight-bearing lunge measure of ankle dorsiflexion. Australian Journal of physiotherapy, 44(3), pp.175-180.
- O'Shea, S. and Grafton, K., 2013. The intra and inter-rater reliability of a modified weight-bearing lunge measure of ankle dorsiflexion. Manual therapy, 18(3), pp.264-268.
- Cejudo et al, 2014. A simplified version of the weight-bearing ankle lunge test: Description and test–retest reliability. Manual therapy, 19(4), pp.355-359.
Therefore, could the authors write how to adjust the angle of the knee during the test and retest please?



Some comments:

1- Line 2, I think the title is general, so it’s could be “Reliability of Android smartphone to measure the range of motion of ankle dorsiflexion” or smartphone application or…
2- Line 25-29, the authors should mention the application or the software name in the Methods
3- Line 66, “Several different measurement…” should put in new Paragraph
4- Line 74, the cited “Williams et al. (2015) not mention in references
5- Line 130, in results “Descriptive…...in Table 1” should be after line 93
6- Line 221, cited Williams et al. (2013) is the same (Williams, Caserta &Haines, 2013)?
7- Lines 220-227, these results for left leg or right leg or both?
8- In conclusions and depending on the results would you recommended for adoption the goniometer application Spirit Level Plus to measure the ankle ROM?

---

## Round 0.2 · Minor Revisions

The paper can be potentially accepted. A reviewer still has some comments. Please revise and give a one-to-one response.

·

Basic reporting

The updated manuscript is clearer and more 'relevant' in its literature use.

Experimental design

This is a reliability study reviewing outcomes from a smart-phone App when used by different people over different days. The test-retest design answers the relaibility but does not determine if the measure is true (e.g. validity of the device).

Validity of the findings

Appear sound

Additional comments

Thank you for your attention to the previously raised issues with this manuscript. I hope you agree that it is a more robust article for the effort.

I am surprised at the author's decision to not test the App against known angles (or compare with a goniometre) and calculate validity. This would take less than an hour to conduct/analyse and would give the study credibility. As it stands, the authors have taken considerable effort to determine that a phone App can measure the same outcome (moderately at least) when used by different people but there is nothing to show that it measures correct angles.

The other concern (minor) is that there is still not enough information regarding the App used for readers to understand how they access it. Can the authors please indicate if the App comes pre-installed in Android phones or requires downloading separately (and if so, is there a cost involved) and credit the developers please. I note the third reviewer also commented on this.

Reviewer 2 ·

Basic reporting

No comment.

Experimental design

No comment.

Validity of the findings

No comment.

Additional comments

I have no further comments on this article. The authors have thoroughly revised the manuscript according to the reviewers comments and suggestions.

Reviewer 3 ·

Basic reporting

Non

Experimental design

Non

Validity of the findings

Non

Additional comments

Most of the comments have been corrected
Thank you

---

## Author Rebuttal · Round 0.2

University of Ljubljana
Faculty of Health Sciences
Zdravstvena pot 5
1000 Ljubljana
Slovenia

Ljubljana, 19.6.2021

Dear editor,

We would like to thank you the reviewers for their comments and suggestions. Their comments are much appreciated. We have addressed their comments and suggestions below and marked the changes as required also in the manuscript. The review comments are in blue and italics, while our response is in bold.

We believe that the manuscript is now suitable for publication in Peer J.

Kind regards,

Renata Vauhnik

On the behalf of all authors

*Reviewer 1 (Helen Banwell)*
*Comments for the Author*
*Thank you for the opportunity to review the manuscript "Reliability of the functional ankle dorsiflexion range of motion measurement in young adults". The article is well written, however there are some concerns that are prohibiting the recommendation for publication at this point.*
*Major concern: the measure capabilities of the app are not being compared to standard practice or 'gold standard' measures (e.g. analogue or digital inclinometers). The reliability of the app is not in question but the authors need to be very clear that this does not guarantee the outcomes are valid. The app could consistently measure incorrectly. Determining validity of the app against a known standard is strongly recommended prior to continuing for publication. This can be done comparing outcomes of the app to known angles (e.g. compare the app to a digital inclinometer and 0 degrees, 15 degrees and 45 degrees using solid and immovable reference points, such as inclinations on ramps).*

**AUTHORS' RESPONSE: Thank you for your comment and your suggestion. Since the purpose of this study was the reliability, validity was not studied. We made this clearer throughout the manuscript. See line 42: "**Further work is required to determine the validity of the mobile goniometer application Spirit Level Plus." **Line 193: "**The validity of the mobile goniometer application Spirit Level Plus has not been studied." **Line 289:** "Further work is required to determine the validity of the mobile goniometer application Spirit Level Plus,…"

*Second recommendation would be to reduce the use of abbreviations, particularly non-standardised abbreviations such as ADROM. It introduces a risk of confusion, breaks sentence flow for the reader and rarely has an adequate effect on word count to be of value. I would recommend the abbreviation be replaced with ankle dorsiflexion ROM throughout. Furthermore, it may be prudent to identify the measure as weight bearing (e.g. weight-bearing ankle dorsiflexion ROM).*

**AUTHORS' RESPONSE: The abbreviation (ADROM) has been removed from the manuscript and has been replaced as suggested. We have also identified the measure as weight-bearing ankle dorsiflexion ROM.**

*Final issue is that much of the literature cited in the background and discussion is > 5 years old. Many newer publications review Android and IOS apps in adults and children. It would be good to update some references, particularly as the*

*Tilt-meter app, which is mentioned several times throughout the manuscript, was decommissioned in 2018.*

**AUTHORS' RESPONSE: Thank you for your comment and your suggestion of the literature. References were updated. See line 77:** "Some authors used special devices (Jones et al., 2005; Watson, Boland & Refshauge, 2008; Morales et al., 2016; Munteanu et al., 2009) or mobile applications of goniometers (Banwell et al., 2019; Goose et al., 2021; Williams, Caserta & Haines, 2013; Vohralik et al., 2015)." **Line 87: "**Two other studies (Gosse at al., 2021; Banwell et al., 2019), investigating the use of Apple IOS based goniometer applications for weight-bearing ankle dorsiflexion ROM measurement, determined moderate to excellent reliability and excellent validity." **Line 149:** "Similar procedure has been used by other researchers (Bennell et al., 1999; Burns & Crosbie, 2005; Rose, Burns & North, 2010; Williams, Caserta & Haines, 2013, Banwell et al., 2019; Goose et al., 2021)." **Line 272:** "Minimal detectable change in our study was higher (5.,0° - 6.,5°) as compared to the results of Williams et al. (2013) (2.,2° - 4.,4°), Gosse et al. (2021) (2.10 – 5.7) and Banwell et al. (2019) (2.4 – 5.0)."

*Abstract*
*Including the number of measures made and more information on the participant cohort would benefit.*

**AUTHORS' RESPONSE: We included the number of measures made and added the information on the participant cohort. Line 29:** "32 young, healthy subjects participated in the study and were measured in four sessions by two examiners. One measurement was taken on each ankle in every session. Eight measurements were taken from each participant. A total of 256 were taken from all the participants."

*Background*
*Good rationale for the need to measure weight-bearing ankle dorsiflexion ROM is given. Further breadth of referencing could be done for the last section where it is currently limited to Williams et al., for using IOS based measures given many other studies have investigated ankle dorsiflexion – e.g:*
*IOS measures of weight bearing ankle dorsiflexion:*
*Banwell, H.A., Uden, H., Marshall, N. et al. The iPhone Measure app level function as a measuring device for the weight bearing lunge test in adults: a reliability study. J Foot Ankle Res 12, 37 (2019).*
*https://doi.org/10.1186/s13047-019-0347-9*

*Carlos Balsalobre-Fernández, Natalia Romero-Franco & Pedro Jiménez-Reyes (2019) Concurrent validity and reliability of an iPhone app for the measurement of ankle dorsiflexion and inter-limb asymmetries, Journal of Sports Sciences, 37:3, 249-253, DOI: 10.1080/02640414.2018.1494908*

**AUTHORS' RESPONSE: The suggested references were included, except the second study by Banwell et al (2019) since their measurement procedure is different and thus not comparable to the one used in our study. Line 77:** "Some authors used special devices (Jones et al., 2005; Watson, Boland & Refshauge, 2008; Morales et al., 2016; Munteanu et al., 2009) or mobile applications of goniometers (Banwell et al., 2019; Goose et al., 2021; Williams, Caserta & Haines, 2013; Vohralik et al., 2015)." **Line 87:** "Two other studies (Gosse at al., 2021; Banwell et al., 2019), investigating the use of Apple IOS based goniometer applications for weight-bearing ankle dorsiflexion ROM measurement, determined moderate to excellent reliability and excellent validity." **Line 149**: "Similar procedure has been used by other researchers (Bennell et al., 1999; Burns & Crosbie, 2005; Rose, Burns & North, 2010; Williams, Caserta & Haines, 2013, Banwell et al., 2019; Goose et al., 2021). **Line 272:** "Minimal detectable change in our study was higher (5.0° - 6.5°) as compared to the results of Williams et al. (2013) (2.2° - 4.4°), Gosse et al. (2021) (2.10 – 5.7) and Banwell et al. (2019) (2.4 – 5.0)."

*This section would also benefit from further rationale on why targeting android apps is important. It erroneously states that there are no publications reporting reliability of Android smart phone apps, which is incorrect (see below) – however the argument could be make that the majority of papers published within the last 5 years are all using Apple IOS based goniometers, and those using Android apps are not measuring dorsiflexion, e.g:*
*Motaz Abdalla Alawna, Bayram H. Unver, Ertugrul O. Yuksel; The Reliability of a Smartphone Goniometer Application Compared With a Traditional Goniometer for Measuring Ankle Joint Range of Motion. J Am Podiatr Med Assoc 1 January 2019; 109 (1): 22–29. doi: https://doi.org/10.7547/16-128*

**AUTHORS' RESPONSE: As pointed out by you, most of the papers published within the last 5 years are all using Apple IOS based goniometers, and those using Android apps are not measuring dorsiflexion. Motaz et al (2019), mentioned above, were measuring dorsiflexion using the Andorid app but in a non-weight-bearing position (supine). This paragraph was re-written. Line 92:** "Although there are several studies, reporting the reliability of the weight-bearing ankle dorsiflexion ROM measurement using Apple IOS based

goniometer applications, evidence of the reliability of a mobile goniometer application using an Android smart phone for weight-bearing ankle dorsiflexion ROM measurement is limited. Therefore, the aim of our study was to determine inter-rater and intra-rater reliability of the weight-bearing ankle dorsiflexion ROM measurement with the flexed knee, using a mobile goniometer application Spirit Level Plus installed on an Android smartphone."

*Methods*
*Could the authors begin this section with a study design description.*

**AUTHORS' RESPONSE: We began the section with a study design description as recommended. Line 102: "**This study followed a test-retest design to determine intra-rater and inter-rater reliability of the weight-bearing ankle dorsiflexion ROM measured by an experienced physiotherapist and a physiotherapy student.**"**

*Participants: It would be good to include the age, height and body weight range of participants and offer some indication of where they were recruited from (e.g. a convenience sample of university students) and how recruitment was conducted..*
*Please indicate if inflammatory joint conditions were also an exclusion criteria.*

**AUTHORS' RESPONSE: Age, height and body weight range of participants are described in the manuscript (line 112 – 113) and in the Table 1. As you have correctly assumed, we used a convenience sample of healthy young subjects. They were recruited during the research leaflet prepared by the first author. Inflammatory joint conditions were also an exclusion criteria. See line 109: "** The inclusion criteria for participation in the study was absence of musculoskeletal injuries in the lower limbs or other disorders of the neuromuscular system, including inflammatory joint conditions, in the last six months prior to the measurements. 32 participants (23 women (72%) and 9 men (28%), age 20.9 ± 1.7 years, body height 170.7 ± 8.1 cm, body mass 66.2 ± 11.6 kg), who signed an informed consent, participated in the study. Research recruitment leaflets were used for participants recruitment."

*Examiners: Raters need to be explained further. Does the experienced physiotherapist use this measure frequently in practice and had the final year physiotherapy student being instructed on how it was conducted prior to the study?*

**AUTHORS' RESPONSE: The required information has been added to the manuscript. The experienced physiotherapist used this measure regularly in her practice and the physiotherapy student has been instructed prior to the study how to perform the measurements. Line 119:** "The measurements were conducted by a physiotherapist with 16 years of clinical experience in the field of musculoskeletal physiotherapy, who has been using this measurement procedure regularly, and a final year physiotherapy student, who has been instructed how to perform the measurements prior to the study. Both examiners are right-handed."

*Did they pilot the study first and refine the protocol?*

**AUTHORS' RESPONSE: Prior to the main data collection, the pilot study was conducted to refine the protocol. Line 122:** "Prior to the main data collection, the pilot study was conducted to refine the protocol."

*Measurement of the functional ADROM: How was the measurement demonstrated to all participants, as a group on one on one.*

**AUTHORS' RESPONSE: The measurement was demonstrated to all participants on one on one. Line 126: "**The measurement procedure was demonstrated to all participants on one on one."

*Given the physiotherapist measured first each time, does this introduce potential order effects (e.g. increasing available ROM due to stretch from the first measure)?*

**AUTHORS' RESPONSE: Examiners performed the first and the second measurements on the different days and therefore the influence of the stretch from the first measure could not occur. Line 130:** "Examiners performed the first and the second measurements on different days, therefore the stretch from the first measurement could not influence the result of the second measurement."

*Can the authors describe the timing and protocol between raters, e.g. could the person rest between? And how were the measures collected (on paper?)*

**AUTHORS' RESPONSE: See our comment above. This was also added to the manuscript. Line 136:** "The measurements were collected on paper separately, so the examiners were blinded to each other's measurements."

*Did the experienced physiotherapist complete and record measures in the presence of the student (or vice versa)?*

**AUTHORS' RESPONSE: See our comment above**

*I also question if 32 people is a large enough sample to ensure that raters don't have memory of previous results.*

**AUTHORS' RESPONSE: See line 134:** "Due to a rather large sample size, and the fact that both sides were being measured, it was impossible for the examiners to memorise the results."

*Also, can the authors discuss the limitations of 'double dipping' when you use both legs on the one person (Menz, 2004)*
*Menz, H. (2004). Two feet, or one person? Problems associated with statistical analysis of paired data in foot and ankle medicine. Foot, 14, 2 - 5.*

**AUTHORS' RESPONSE: We agree, and we are aware that the correlation of paired data (ie, two legs of a person) is an issue which might result in spurious findings, however our analysis (all the calculations) was done separately for left and for right foot for reliability within or between examiners. Each of the results refers to only one foot of a person therefore no »double dipping« was done in our case.**

*Further information on the App, e.g. reference to its developers and which operating systems are required etc., and is it free to download or involve a subscription etc., .*

**AUTHORS' RESPONSE: Added. Line 147:** "Application is free to download, no subscription is needed. Android 1.6+ operating systems are required."

*Statistical procedure*
*How was all the data managed? It would benefit to include the participant characteristic management here (e.g. Descriptive statistics were used to describe participants...).*

**AUTHORS' RESPONSE: Descriptive statistics of participants is presented in Table 1.**

*Can the authors rationalise the use of ICC random rather than two-way mixed effect (i.e. Model 2, 1 rather than Model 3, 1)?*

**AUTHORS' RESPONSE: ICC (2,1) was used in order to be able to generalize the results (ie, agreement, Shrout and Fleiss (1979)) wider than to only the two particular examiners (when considering reliability between examiners). When considering reliability within examiners, ICC (2,1) is a commonly used method.**

*The authors could consider using Bland Altman Plots to ensure the limits of agreement of raters is within an acceptable level. This could also be used to explore validity if the app is compared with a digital goniometer as recommended.*

**AUTHORS' RESPONSE: We agree that the results could be visually represented with Blant Altman plot. 95% limits of agreement in our case would be constructed with the reported MDC. Bland Altman plots as Figure 2 is included in the manuscript (Line 173).**

*Results*
*Please change the word smallest to minimum in line 134.*

**AUTHORS' RESPONSE: Corrected. Line 172:** "Minimum detectable change was determined at least 4.99 degrees. The results are shown in Table 3."

*Discussion*
*You could incorporate this paper into your discussion given it included a final year physiotherapy student and a novice (parent). It would offer you a clearer comparison that the Williams study.*
*Gosse G, Ward E, McIntyre A, Banwell HA. 2021. The reliability and validity of the weight-bearing lunge test in a Congenital Talipes Equinovarus population (CTEV) PeerJ 9:e10253 https://doi.org/10.7717/peerj.10253*

**AUTHORS' RESPONSE: Thank you for your suggestion. This study was added to the Discussion. Line 183: "**The results are comparable to the results of Gosse et al. (2021), who determined moderate to excellent reliability of the mobile goniometer application iPhone level, with ICC ranging from 0.68 to 0.90." **Line 208**: "Interestingly, the intra-rater reliability of a clinician in training and a novice rater was higher (ICC 0.90) when using the iPhone app then the intra-rater reliability of an experienced clinician (ICC 0.75) (Gosse et al., 2021)." **Line**

**272: "**Minimal detectable change in our study was higher (5.0° - 6.5°) as compared to the results of Williams et al. (2013) (2.2° - 4.4°), Gosse et al. (2021) (2.10° – 5.7°) and Banwell et al. (2019) (2.4° – 5.0°)."

*Sentence beginning line 206 requires a reference.*

**AUTHORS' RESPONSE: Added. Now line 253:** "Those who used a gravity goniometer or a mobile goniometer application measured the tibial inclination on different locations (superior to the posterior calcaneal tuberosity (Dickson et al., 2012; Williams, Caserta & Haines, 2013, Banwell et al., 2019; Goose et al., 2021), on the lateral (Cejudo et al., 2014) or anterior (Bennell et al., 1998; Dickson et al., 2012; Vohralik et al., 2015) part of the tibia, at different heights)."

*Please replace comma's for full stops when reporting the Minimal Detectable change (Lines 220 to 228).*

**AUTHORS' RESPONSE: Replaced.**

*References:*
*Evans, Rome & Peet, 2012 is not in the reference list. Please review to ensure all other references are included.*

**AUTHORS' RESPONSE: Corrected and included.**
* * *
*Reviewer 2 (Anonymous)*
*Basic reporting*
*The manuscript was very clear, easy to read and was conducted well. Considering the language, I just found few suggestions and typing errors. In lines 79-83 (at the end of introduction) I would suggest to make two different sentences instead of one, and in the heading of table 2 there is a small spelling mistake.*

**AUTHORS' RESPONSE: Corrected as suggested.**

*In the Table 1, the Range is somewhat confusing, I would suggest to leave that out or show the range more understandable, like 19-27.*

**AUTHORS' RESPONSE: Corrected.**

*Experimental design*
*Fine, no comments to add.*
*Validity of the findings*
*Interesting study, no comments to add.*
* * *
*Reviewer 3 (Anonymous)*
*Basic reporting*
*no comment*
*Experimental design*
*Some comments in general section*
*Validity of the findings*
*no comment*
*Comments for the Author*
*I have read the article ID 58796 under title "Reliability of the functional ankle dorsiflexion range of motion measurement in young adults".*
*In this article, the authors reported their work using a mobile goniometer application Spirit Level Plus from Android smartphone and recruited 32 subjects in study. The overall, article writing is good and the results analyses are reasonably thorough. But although, the authors used new application to measure the ankle dorsiflexion, the study is still preferable that if the authors can take the following comments into consideration in preparing the final version.*

*I recommend that the authors make changes related to the comments below.*

*Reviewer comments:*

*-Figure 1, shown (weight bearing lunge test) but at the same time what the degree of knee flexion during test. Maybe the same subject when did re-test the degree of knee is different so I think that the best position is for the knee to be parallel to the foot, and the distance (to the nearest 0.lcm) from the end of the big toe to the wall using a tape measure on the floor and, for more information could see references:*
*- In line 297(Konor et al.);*
*- Bennell et al, 1998. Intra-rater and inter-rater reliability of a weight-bearing lunge measure of ankle dorsiflexion. Australian Journal of physiotherapy, 44(3), pp.175-180.*
*- O'Shea, S. and Grafton, K., 2013. The intra and inter-rater reliability of a modified weight-bearing lunge measure of ankle dorsiflexion. Manual therapy, 18(3), pp.264-268.*

*- Cejudo et al, 2014. A simplified version of the weight-bearing ankle lunge test: Description and test–retest reliability. Manual therapy, 19(4), pp.355-359. Therefore, could the authors write how to adjust the angle of the knee during the test and retest please?*

**AUTHORS' RESPONSE: Thank you for comment. Knee flexion angle is important when considering the effect of the musculus gastrocnemius on ankle dorsiflexion (Baumbach et al., 2014), however since the purpose of this study to evaluate the reliability of the weight-bearing ankle dorsiflexion range of motion measurement using a smartphone goniometer application, we did not control the knee flexion in the measurements.**

*Some comments:*

*1- Line 2, I think the title is general, so it's could be "Reliability of Android smartphone to measure the range of motion of ankle dorsiflexion" or smartphone application or…*
**AUTHORS' RESPONSE: Considering your comment and comment from the other reviewer the title has been changed into: "Reliability of the weight-bearing ankle dorsiflexion range of motion measurement using a smartphone goniometer application."**

*2- Line 25-29, the authors should mention the application or the software name in the*

**AUTHORS' RESPONSE: Added. Line 25:** "The purpose of the study is to assess the intra-rater and the inter-rater reliability of the weight-bearing ankle dorsiflexion range of motion measurement with a flexed knee using a smartphone application Spirit Level Plus installed on an Android smartphone."

*Methods*
*3- Line 66, "Several different measurement…" should put in new Paragraph*

**AUTHORS' RESPONSE: Corrected.**

*4- Line 74, the cited "Williams et al. (2015) not mention in references*

**AUTHORS' RESPONSE: This was a typing error. It should be written 2013. Now it is corrected throughout the manuscript.**

*5- Line 130, in results "Descriptive......in Table 1" should be after line 93*

**AUTHORS' RESPONSE: Corrected.**

*6- Line 221, cited Williams et al. (2013) is the same (Williams, Caserta &Haines, 2013)?*
**AUTHORS' RESPONSE: Yes, it is the same.**

*7- Lines 220-227, these results for left leg or right leg or both?*

**AUTHORS' RESPONSE: Clarified. Line 281** »In the study by Konor et al (2012), Cejudo et al (2014) and our study, reliability was performed for both legs, while in the study by Williams et al (2013) reliability was performed only for the right leg and in study by Dickson et al (2021) and Bennell et al (1998) reliability was performed only for the left leg."

*8- In conclusions and depending on the results would you recommended for adoption the goniometer application Spirit Level Plus to measure the ankle ROM?*

**AUTHORS' RESPONSE: Since the present study investigated only the reliability of the weight bearing ankle dorsiflexion range of motion measurement using smartphone goniometer, before recommending its use in the clinical and research setting, the validity study should be performed. This was added to the conclusion. Line 289:** "Further work is required to determine the validity of the mobile goniometer application Spirit Level Plus, normative values of ankle dorsiflexion ROM and to determine the differences in ankle dorsiflexion ROM among different age groups and gender, before its recommendation in clinical settings."

---

## Round 0.3 · accepted · Accept

The paper is good now. It can be accepted.